# The Influence of an Integrated Driving on the Performance of Different Passive Heating and Cooling Methods for Buildings

**Ivan Oropeza-Perez**

Department of Architecture, Universidad de las Americas Puebla, Ex Hacienda Sta. Catarina Martir, San Andres Cholula 72810, Puebla, Mexico; ivan.oropeza@udlap.mx

**Abstract:** Passive cooling and heating methods within buildings are studied a lot nowadays. Nevertheless, their performance considering their driving has not been deeply studied. Therefore, the performance of the most common passive heating and cooling methods is analyzed in this document. The methods are divided into three categories: operable, semi-operable, and not operable. They are studied under different conditions of operation in order to estimate their performance in terms of indoor temperature increase/decrease in a single dwelling. The study is carried out with the thermal simulation program EnergyPlus, using a dwelling in Mexico City as a case study, which is validated with literature that studied passive methods for similar climates. Furthermore, for an integrated driving, four features of operation of the passive methods are considered: mobility, maintenance, assembly, and consumables. The results show that a correct use of these features of driving might achieve a significant temperature drop in the case of cooling and a significant indoor temperature increase in the case of heating. This is reflected in a considerable amount of energy saving compared to a conventional heating/cooling heat-pump system running under regular conditions, which is taken as a reference of consumption. Thereby, it is concluded that the proper usage, considered here as the correct application of the four features of operation mainly by the occupants, might have a high influence in their performance of increase/decrease of the indoor temperature. Thus, it is highly recommended to follow up their performance once installed and not to suppose an optimal performance ever after.

**Keywords:** passive cooling; passive heating; integrated driving; effectiveness of operation; features of operation

---

## 1. Introduction

The current situation in the world obligates new and innovative manners of providing the everyday growing energy necessary to keep the levels of production and way-of-life of humans. Moreover, it is well-known that global warming, closely related to energy activities, is a threat that has to be addressed immediately by the current and the future generations [1,2].

One of the manners of countering climate change is related to the efficient use of energy. In this context, the building sector is a key target due to its share of the world's energy use (40%) [3]. Out of this consumption, one of the most consuming activities within the buildings is the space conditioning, either for cooling or for heating [4].

In this sense, passive cooling and heating strategies were developed to achieve thermal comfort without spending great amounts of energy and other resources. In all cases, the methods decrease or increase the indoor temperature, depending on their original purpose [5–14]. Nevertheless, sometimes these methods are not capable of reaching the same temperature drop, in the case of

cooling, and temperature increase, in the case of heating, as active methods such as heat pumps, air-conditioning (AC), or radiation systems [15].

Concerning the passive methods, a review shows that they can be divided according to their positive or negative heat-flow path onto the building heat balance, when this one does not have any active heating or cooling system (free-running building). These paths can be generally classified as the following: internal heat gains, heat transfer through the envelope, and heat transfer between the indoor and the outdoor air (see Figure 1).

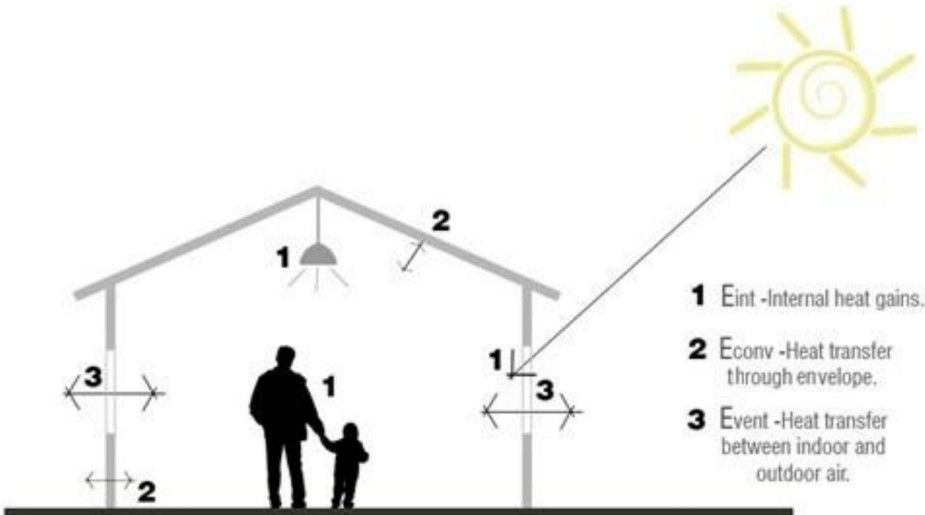

**Figure 1.** Different heat-flow paths within a free-running building, derived from authors' elaboration.

In general, the heat transfer is by radiation for the internal heat gains, conduction for the heat transfer through the envelope, and convection for the heat transfer between the indoor and outdoor air. Nevertheless, the heat transfer manner might vary for some passive methods. Thereby, the most studied and used passive methods can be gathered as follows (Table 1).

**Table 1.** Classification of passive methods by heat flow.

| Heat Flow | Passive Method [16–52] |
|---|---|
| Internal heat gains | Passive solar gain <br> Shading systems |
| Heat transfer through the envelope | Thermal insulation <br> Double-glazed opening <br> Phase change materials <br> Passive cooling shelter <br> Heat sink <br> Thermal capacity <br> Radiant heat barrier |
| Heat transfer between the indoor and the outdoor air | Infiltration control <br> Eco-evaporative cooling <br> Natural ventilation <br> Solar-assisted AC |
| Multiple heat transfer means | Intelligent facade |

Table 1 presents the most studied and used passive cooling and heating methods. Out of the fourteen presented methods, and according to the literature review, passive solar gain and infiltration control aim to heat up the space. Intelligent facade, thermal insulation, and double-glazed opening have both heating and cooling as their objectives, while the remaining nine methods are generally used for

cooling [53,54]. Furthermore, from the literature review, it is seen that, out of the 37 reviewed documents, 35 present studies under similar climate conditions, i.e., group C and D of the Köppen–Geiger climate classification system [3]. Hence, a comparison between the reviewed methods and their use in a similar climate could be carried out in this document.

Furthermore, one of the most influencing factors for the proper performance of passive methods is operation, understood as the handling that the passive methods need in order to increase or decrease the indoor temperature, according to the case. For instance, to keep the indoor heat gain is necessary to close all the openings; to block the sunbeams the blinds must be shut; or to reflect the solar radiation the building should be white-painted. These handlings could be in a short, medium, or long period of time, depending on the method and its features.

Nevertheless, to the best of our knowledge, no study deals with the operation features and their influence on the passive methods' performance. In this sense, the occupants are only seen as a response of the energy consumption performance (not only by passive and active methods but also by others such as appliances and illumination), depending on their necessities and comfort levels [55–63]. The passive methods are generally considered static, where, once installed, there is not a driving beyond a certain amount of maintenance and when they break down, if it is the case. Although some passive methods are dynamic and can be controlled automatically or manually by the occupants [64–70], there are other features to take into account if the passive methods are considered with an integrated driving.

Thus, in this document, the aforementioned passive heating and cooling methods are analyzed in terms of their operation characteristics. A case study in Mexico City is presented in the document, considering that this city is suitable to compare to places with similar climates when assessing the reviewed passive methods, as was mentioned before.

The purpose of this is to determine the extent of drop/increase of the indoor temperature according to the corresponding handling of each passive method and to develop a guidance of operation in order to optimize their performance. Nonetheless, the geographical scope of this study is limited to locations with warm-to-hot conditions that present warm summers and cold winters (groups C and D of the Köppen–Geiger climate classification system).

### 1.1. Classification by Time between Driving

The different passive methods can be gathered according to one type of performance, namely, the period of time between one driving and another. In this case, a driving is understood as any kind of maintenance, operation, fixing, or other action that the building user applies to the passive method in order to correct or modify its performance. The classification of the passive methods is carried out by taking into account the amount of time between each driving, as shown in Table 2.

**Table 2.** Classification of passive methods by time between driving.

| Classification | Time Between Driving |
|---|---|
| **Operable** | ≤1 week |
| **Semi-operable** | > 1 week ≤ 1 year |
| **Not operable** | >1 year |

Considering the fourteen passive methods available (two for heating, nine for cooling, and three for both heating and cooling) as the most used nowadays within dwellings [62,63], a classification of these methods by their time between driving can be carried out. Table 3 shows this classification, along with the main purpose of each method, i.e., heating, cooling, or both. For further information about the characteristics and the time between driving of the passive methods, please refer to [62,63].

**Table 3.** Classification of the passive cooling and heating methods by time between driving.

| Passive Method | Operation Condition |
| --- | --- |
| **Heating** | |
| Passive solar gain | Operable |
| Infiltration control | Semi-operable |
| **Cooling** | |
| Shading system | Operable |
| Phase change material | Not operable |
| Passive cooling shelter | Semi-operable |
| Heat sink | Semi-operable |
| Thermal capacity | Not operable |
| Radiant heat barrier | Semi-operable |
| Eco-evaporative cooling | Operable |
| Natural ventilation | Operable |
| Solar-assisted AC | Operable |
| **Heating/Cooling** | |
| Thermal insulation | Not operable |
| Intelligent facade | Semi-operable |
| Double-glazed opening | Operable |

From Table 3, it can be seen that only three methods are considered to be not operable, while the other eleven need some kind of driving before one year passes after installation.

*1.2. Features of Operation*

The operation of the passive methods is also classified by their features. These characteristics are stated in this document as the following:

- *Mobility.* Related to the self-adjustment of the passive method for maximizing its performance according to the external and internal conditions; it is also defined as the control of the method, e.g., the opening of two opposite doors for cross ventilation.
- *Maintenance.* Understood as all physical action applied to the passive-method mechanism in the long-term for increasing its performance. For instance, to clear the window glass for passive solar heating.
- *Assembly.* Stated with regard to its placement and displacement according to the status of the passive method. For example, the placement of an external window shutter to keep the indoor heat.
- *Consumables*. Defined as the resources used to perform the passive methods. One example of this is the water used to decrease the indoor temperature by applying eco-evaporative cooling.

Thereby, Table 4 can be constructed. In this table, the heating and cooling passive methods are shown, along with their main operation features. As it can be seen in Table 4, apart from the three not-operable methods, which only need a certain level of maintenance, all passive methods have at least two features of operation that imply the driving by the occupants. The variation of these features makes an integrated driving, which is related to their performance, which is hereby called the effectiveness of operation of the methods.

**Table 4.** Classification of passive cooling and heating methods by operation features.

| Passive Method | Operation Features |
|---|---|
| **Heating** | |
| Passive solar gain | Mobility, maintenance |
| Infiltration control | Mobility, maintenance, assembly |
| **Cooling** | |
| Shading system | Mobility, maintenance, assembly |
| Phase change material | Maintenance |
| Passive cooling shelter | Mobility, maintenance, consumables |
| Heat sink | Maintenance, consumables, assembly |
| Thermal capacity | Maintenance |
| Radiant heat barrier | Maintenance, consumables, assembly |
| Eco-evaporative cooling | Mobility, maintenance, assembly, consumables |
| Natural ventilation | Mobility, assembly |
| Solar-assisted AC | Mobility, maintenance, consumables |
| **Heating/Cooling** | |
| Thermal insulation | Maintenance |
| Intelligent facade | Mobility, maintenance, consumables |
| Double-glazed opening | Mobility, maintenance |

## 2. Methodology

### 2.1. Effectiveness of Operation

The effectiveness of operation of the methods could be estimated. This effectiveness is stated as the difference between the indoor air temperature increase/decrease that can be achieved and a previously set indoor temperature of comfort, as shown in Equation (1), developed for this document to obtain a linear comparison between the values:

$$\varphi = 100 - \left( \frac{\left| T_{Comfort} - T_{Passive} \right|}{T_{l-u}} \cdot 100 \right) \tag{1}$$

The effectiveness has a value between 0% and 100%, where the range of thermal comfort is delimited by a lower and an upper threshold, $T_{l-u}$, which is in accordance with international standards, such as ASHRAE 55 and EN 15251 [71]. Nonetheless, it is worth mentioning that the thermal comfort range can vary due to other factors, such as clothing, metabolic rate, or relative humidity, among others, and depends on human factors, such as the psychological and physiological perception [71].

Thereby, for instance, Figure 2 shows the effectiveness of a passive method with a temperature setpoint of 22 °C, considering a thermal comfort range between 18 and 26 °C ($T_{l-u}$ = 4 °C). In this case, if a passive method achieves 22 °C, its effectiveness is 100%. If it achieves 23 °C, its effectiveness would be 75%.

It is worth to mentioning that the effectiveness calculated by Equation (1) is a subjective value that depends on two features: the temperature of comfort and the lower and upper threshold. Both values are taken from adaptive thermal comfort models that are dependent on both physical and human factors [71]. Therefore, the effectiveness of operation is a fixed value to indicate how well a method heats up or cools down a space, but it is not free of accuracy fails and perception errors.

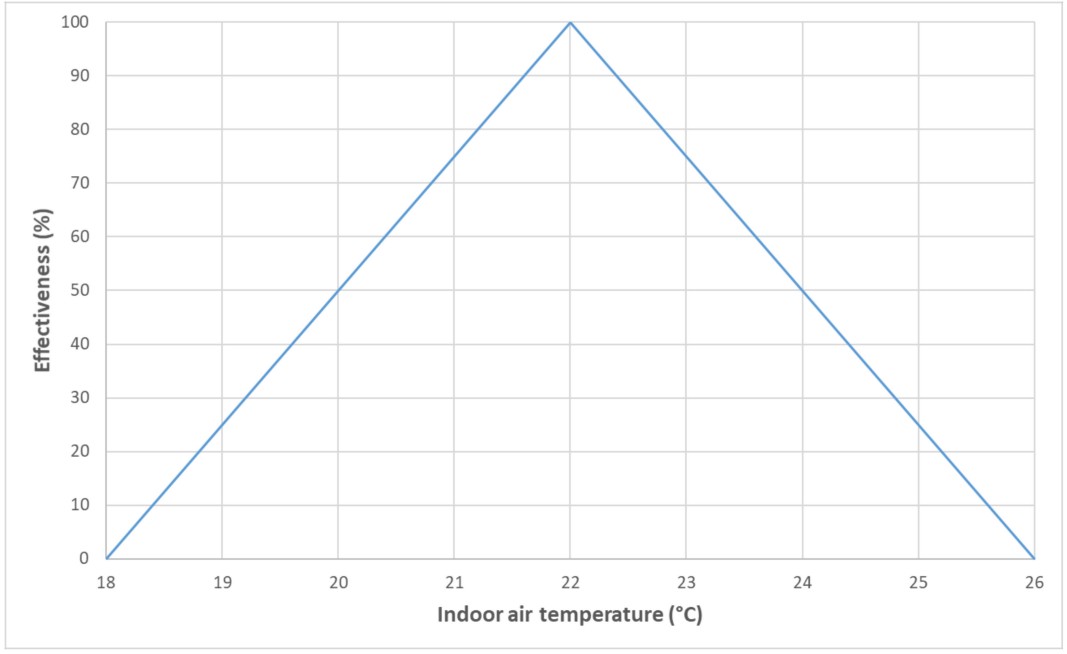

**Figure 2.** Effectiveness of the passive method for a temperature of comfort of 22 °C.

### 2.2. Influence of Performance

Another concept that is introduced in this document is the 'influence of performance', which is defined as the influence that the four aforementioned operation features have upon the performance (temperature increase/decrease) of the passive methods. For defining a value of this influence of performance, it is considered that the operation features have an index of influence, shown in Table 5. It is considered that the four indexes have 100% influence over the decrease or increase of the indoor temperature. Furthermore, the values were weighted according to the literature review of the passive methods, where these four operation features are presented with different performances, if they are applied. The index of influence of each operation feature was estimated based on the temperature increase or decrease presented in the reviewed documents.

**Table 5.** Value of the influence index of the operation features.

| Operation Feature | Influence Index (%) |
|:---:|:---:|
| Mobility | 35 |
| Maintenance | 15 |
| Assembly | 30 |
| Consumables | 20 |

Thereby, depending on the number of operation features, the passive method can decrease or increase the indoor temperature at a maximum value (influence of performance of 100%). If the passive method does not have any operation feature, it is considered that the influence of performance is 0%, i.e., there is no variation in the indoor temperature. Figure 3 can be displayed showing the influence of performance of the fourteen passive heating and cooling methods.

In Figure 3, it can be noticed that the only method that can be 100% controlled is eco-evaporative cooling, whereas phase change material and thermal capacity have only up to 15% of the indoor temperature variation when their only operation feature (maintenance) is applied.

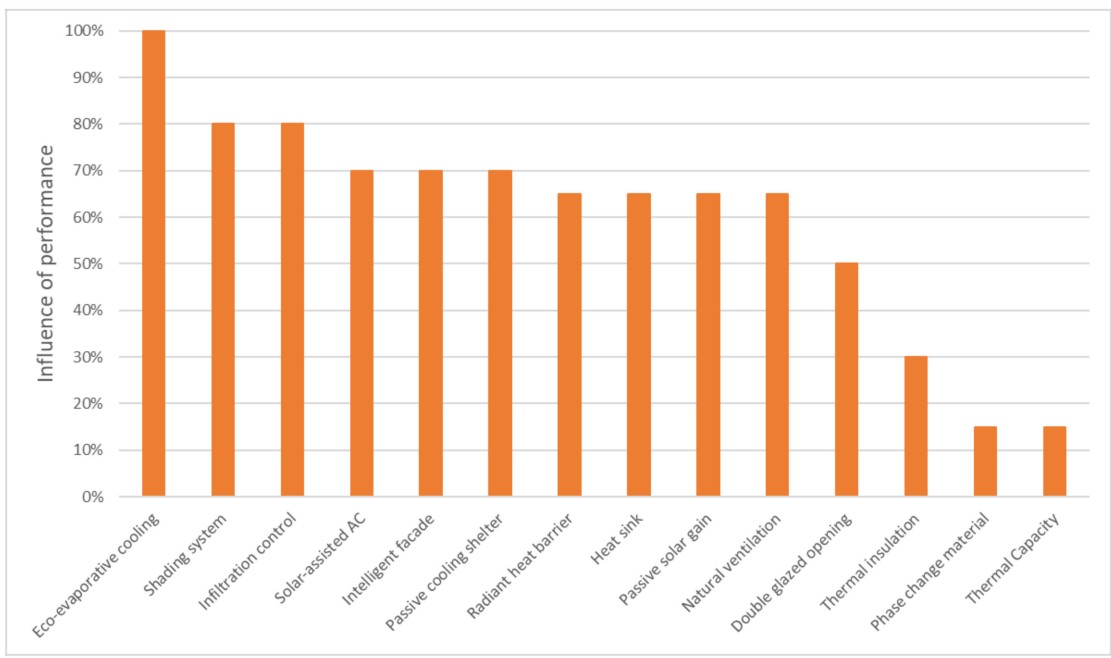

**Figure 3.** Influence of performance of the passive methods.

## 3. Results

### 3.1. Passive Methods' Performance

The performance and therefore the effectiveness of the passive methods is determined by the indoor temperature that each method achieves in particular. For this, the thermal simulation program EnergyPlus [72] is used in order to calculate the indoor temperature by utilizing the different passive methods. When the indoor temperature is calculated, it is compared to the temperature of comfort by using Equation (1) and thus estimating the effectiveness $\varphi$.

### 3.2. Characteristics of the Simulation Modelling

The estimation of the indoor temperature is carried out in an hourly way for the warmest and the coldest day of the year, respectively, taking a dwelling in Mexico City as a case study. This city was chosen because it represents, in a clear manner, a warm cooling season (average maximum outdoor temperature of 30.15 °C) and a cold heating season (average minimum outdoor temperature of 2.75 °C) [73,74], considered as Cwb in the Köppen–Geiger climate classification system. The temperature of comfort was set at 22 °C, as is established for Mexico, with a thermal comfort range between 18 and 26 °C [71].

One single dwelling is studied in the document. The modelled dwelling has the most common materials in Mexico (brick walls of 10 cm of width and a concrete roof of 30 cm of width, with four windows of 6 mm clear glass), with an average built area of 120 m$^2$ and a ceiling height of 2.2 m (see Figure 4). Furthermore, the occupancy was stated as a regular home schedule (absences during the day and full occupation during the night), considering four occupants with mechanical heat gains of 1000 W. For the simulation model, it is considered that the dwelling has a backyard and it is placed within a low, dense urban area (see Figure 5), where its external walls are adiabatic with the adjacent surfaces. It is also considered that the south facade is fully sun-exposed.

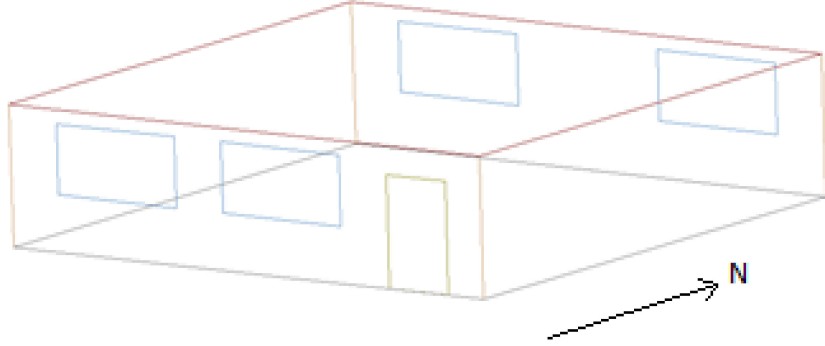

**Figure 4.** Geometry of the simulated dwelling. Source: own elaboration based on EnergyPlus.

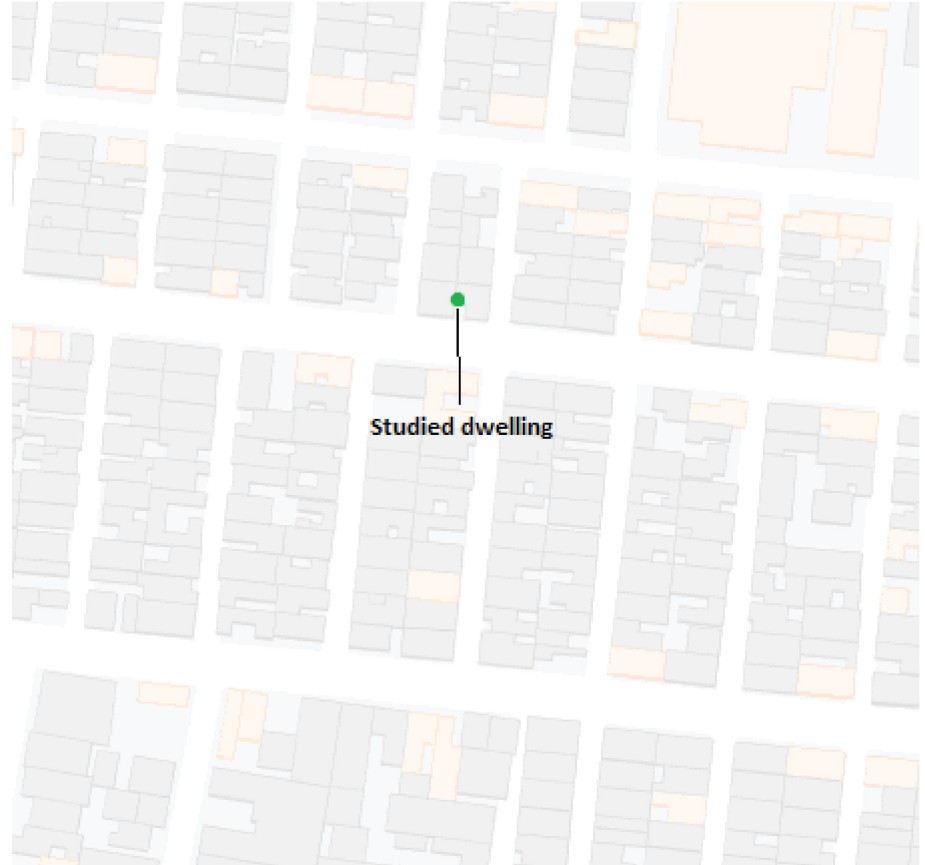

**Figure 5.** Urban location of the simulated dwelling (green dot) [75].

Furthermore, to estimate the hourly indoor temperatures achieved by the respective passive methods within the simulation modelling, different assumptions were considered. For instance, the indoor temperature achieved by radiant heat barriers was calculated by increasing the reflectivity of the surface; the performance of thermal insulation was estimated by including an extra layer of insulation material; or for thermal capacity, the heat capacity of the walls' material was enhanced.

### 3.3. Results of Simulation

Within the EnergyPlus program, the characteristics of the modelling (schedule, materials of construction, geometry, location, etc.) are set as input data. The characteristics of the passive methods are also set as inputs, mainly by changing the physical characteristics of the building (double-glazed windows, radiant heat barriers, controlled natural ventilation, etc.). Using the simulation modelling

for every passive method, Table 6 is constructed, considering that the temperatures were achieved without any kind of control, mobility, or maintenance by the occupants and setting a temperature of comfort of 22 °C.

**Table 6.** Indoor temperature of passive cooling and heating methods without any occupants driving.

| Passive Method | Highest Temperature °C (Heating) | Lowest Temperature °C (Cooling) |
|---|---|---|
| **Heating** | | |
| Passive solar gain | 10.84 | - |
| Infiltration control | 11.89 | - |
| **Cooling** | | |
| Shading system | - | 24.86 |
| Phase change material | - | 22.96 |
| Passive cooling shelter | - | 31.61 |
| Heat sink | - | 23.97 |
| Thermal capacity | - | 24.65 |
| Radiant heat barrier | - | 26.31 |
| Eco-evaporative cooling | - | 31.27 |
| Natural ventilation | - | 24.71 |
| Solar-assisted AC | - | 32.44 |
| **Heating/Cooling** | | |
| Thermal insulation | 15.17 | 28.83 |
| Intelligent facade | 19.20 | 24.80 |
| Double-glazed opening | 11.78 | 25.73 |

In Table 6, it can be seen that none of the methods reach the comfort indoor temperature (22 °C). Therefore, the operation features must be applied in order to achieve thermal comfort with the passive methods.

### 3.4. Validation of the Results

The results obtained by the simulations are compared with previous works that focused on the thermal performance of certain methods [62,76–78]. It is worth mentioning that some passive methods have not been thermally analyzed; thus, there is no data available. Also, it is important to clarify that, in most of the cases, the conditions of construction, occupancy, and others were not exactly the same, and thus it is expected to have certain discrepancies with the simulated results.

Thereby, the comparison of the indoor temperatures can be seen in Table 7. In the table, the highest simulated indoor temperature increase or decrease is taken into account and it is compared to the highest indoor temperature increase or decrease taken from the literature review.

As seen in Table 7, the highest error between the reviewed indoor temperature increase/decrease and the simulated one is the given by passive solar gain (36%). In all cases, the error can be considered as acceptable. Furthermore, since References [62,76–78] are carried out under climate conditions within groups C and D of the Köppen-Geiger classification, and, along with the reviewed passive methods of References [16–52], it is considered that the methods are working under similar conditions; hence, the results can be stated as valid.

**Table 7.** Comparison of the thermal performance of the passive methods.

| Passive Method | Averaged Reviewed Temperature Increase/Decrease (°C) [16–52,62,76–78] | Simulated Temperature Increase/Decrease (°C) | Error (%) |
|---|---|---|---|
| **Heating** | | | |
| Passive solar gain | 7 | 11 | 36 |
| Infiltration control | NA | 10 | NA |
| **Cooling** | | | |
| Shading system | 3 | 3 | 0 |
| Phase change material | NA | 6 | NA |
| Passive cooling shelter | 20 | 15 | 25 |
| Heat sink | 6 | 7 | 14 |
| Thermal capacity | 8 | 7 | 12 |
| Radiant heat barrier | 13 | 9 | 31 |
| Eco-evaporative cooling | 6 | 8 | 25 |
| Natural ventilation | 15 | 14 | 7 |
| Solar-assisted AC | 20 | 15 | 25 |
| **Heating/Cooling** | | | |
| Thermal insulation | 9 | 7 | 22 |
| Intelligent facade | 10 | 8 | 20 |
| Double-glazed opening | 22 | 15 | 25 |

### 3.5. Effectiveness of the Passive Methods

By applying Equation (1) and the results shown in Table 6, Figure 6 can be displayed with the effectiveness of each passive method. It is important to mention that, because in the literature it is generally expressed in this manner, for thermal insulation and double-glazed opening, it is considered the temperature of heating, whereas, for intelligent facades, the temperature of cooling was taken into account.

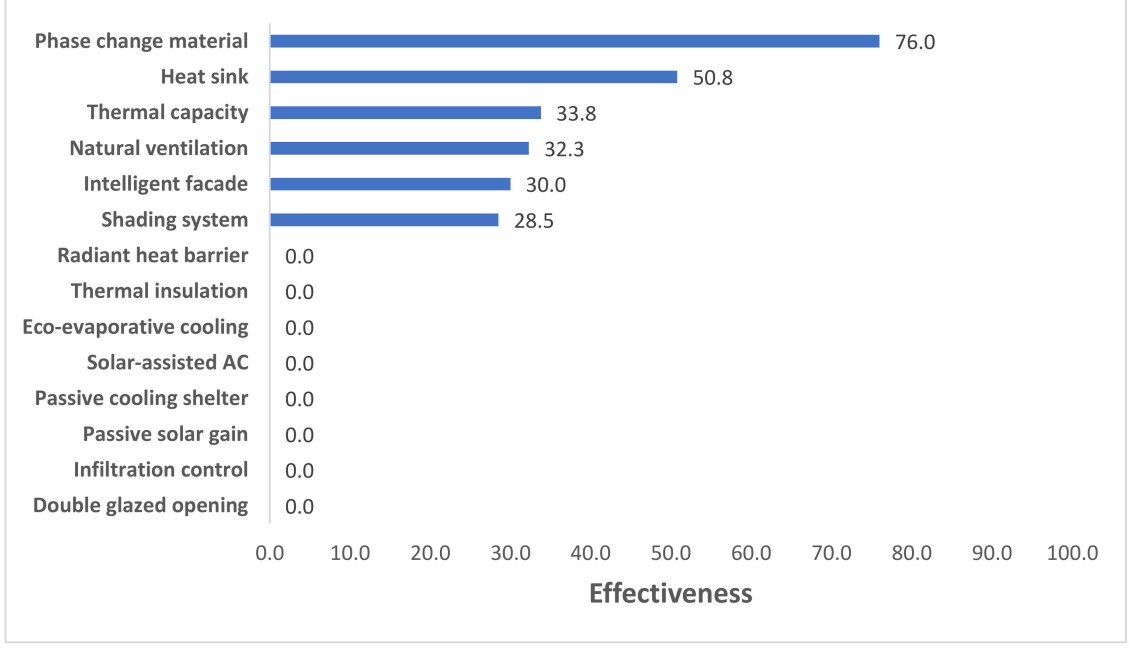

**Figure 6.** Effectiveness of the passive methods for a comfort temperature of 22 °C.

From Figure 6, one can see that eight out of the fourteen passive methods do not accomplish a minimal effectiveness to be considered a correct method for achieving thermal comfort. It can also be seen that, out of the six methods that fulfill the minimal effectiveness, two of them, i.e., phase change

material and thermal capacity, are defined as not operable. These results are good because only one operation feature can be applied upon them (see Table 4); therefore, once they are installed, they do not need driving often.

Regarding the other passive methods, and placing emphasis onto the eight that did not accomplish the effectiveness, the operation features mentioned in Table 4 and Figure 3 are varied until their optimal effectiveness is reached. The operation features are changed in the simulation modelling onto the physical characteristics of the building design and the variation of the occupant schedule. Therefore, results of the indoor temperature with a proper occupants driving can be shown in Table 8. In real life, this change of characteristics to the simulation modeling is seen, for example, as the following:

- *Mobility.* Opening of the windows and doors to enhance natural ventilation. Opening and closing of blinds to either allow or block the direct solar gains. Adapting a double-glazed window to retain or expel heat. Etc.
- *Maintenance.* Cleaning of the windows. Watering of green surfaces. Cleaning of reflective surfaces. Draining of the pipelines of the passive cooling shelter. Etc.
- *Assembly.* Placement of sealing material into cracks of windows and doors to increase airtightness. Placement of more transparent blinds to allow direct solar gains. Etc.
- *Consumables.* Use of water in a swimming pool to create a natural heat sink. Water for eco-evaporative cooling. Working fluid of a solar-assisted AC. Etc.

**Table 8.** Indoor temperature of passive cooling and heating methods with occupants driving.

| Passive Method | Highest Temperature °C (Heating) | Lowest Temperature °C (Cooling) |
|---|---|---|
| **Heating** | | |
| Passive solar gain | 18.72 | - |
| Infiltration control | 18.58 | - |
| **Cooling** | | |
| Shading system | - | 24.60 |
| Phase change material | - | 22.56 |
| Passive cooling shelter | - | 24.05 |
| Heat sink | - | 23.04 |
| Thermal capacity | - | 23.61 |
| Radiant heat barrier | - | 22.74 |
| Eco-evaporative cooling | - | 22.66 |
| Natural ventilation | - | 22.32 |
| Solar-assisted AC | - | 23.44 |
| **Heating/Cooling** | | |
| Thermal insulation | 18.16 | 25.84 |
| Intelligent facade | 19.89 | 24.11 |
| Double-glazed opening | 18.54 | 23.34 |

As mentioned previously, the physical characteristics of the simulation modelling were varied (increasing the thermal transmittance for passive heating, decreasing the thermal absorption for radiant heat barrier, etc.). Thereby, from the indoor temperature values of Table 8 achieved by each passive method, Figure 7 is constructed. One more time, for thermal insulation and double-glazed opening, the achieved temperature of heating was considered, whereas, for intelligent facades, the temperature of cooling was taken into account.

Figure 7 shows that the effectiveness of almost all the passive methods had a considerable improvement by handling one or more operation features (mobility, maintenance, assembly, and consumables).

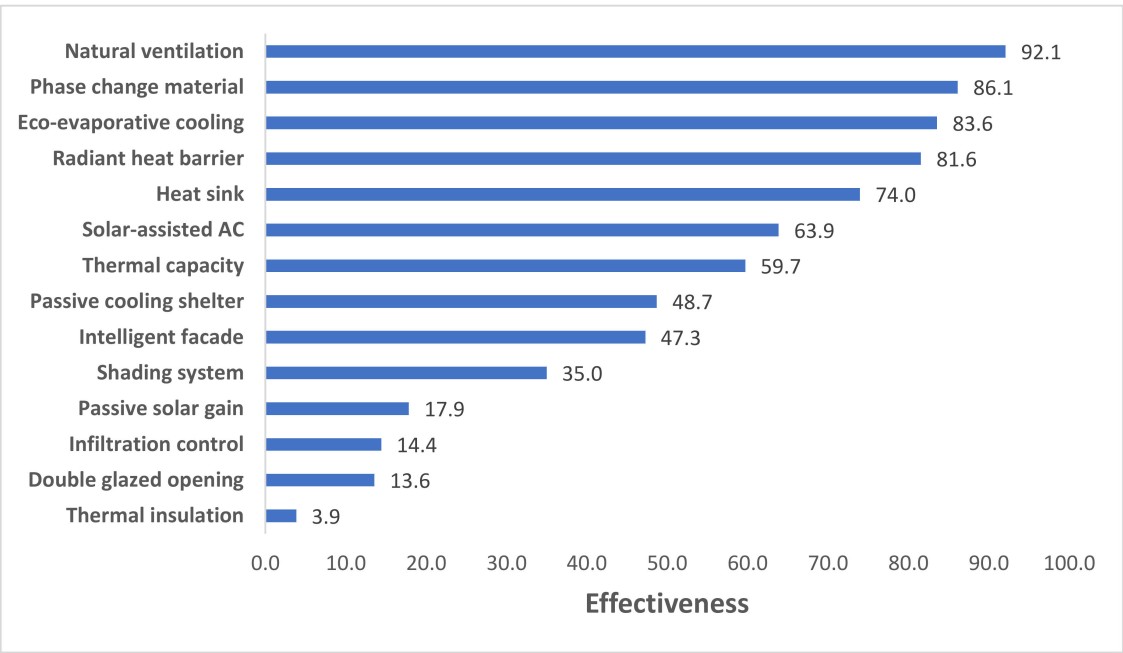

**Figure 7.** Effectiveness of the passive methods for a comfort temperature of 22 °C and having a driving by the occupants.

Analyzing the results, it can be seen that the highest improvements of the effectiveness were achieved by the passive methods that have high percentages of the influence of performance (cf. Figure 3), i.e., eco-evaporative cooling, solar-assisted AC, radiant heat barriers, passive cooling shelter, and natural ventilation. These passive methods have the advantage of giving users control over them (mobility), along with other operation features, such as often requiring maintenance and allowing for the use of consumables (electricity, water, refrigerant, etc.). Furthermore, the classification by operation conditions (cf. Table 3) indicates that they are considered to be operable or semi-operable methods.

Moreover, the three passive methods considered as not operable reached low improvements. This is explained for the low values of influence of performance (cf. Figure 3) that these methods present and which are reflected in a certain maintenance over a period of more than one year.

*3.6. Energy Saving Estimation*

Taking as the baseline a heat pump with an average coefficient of performance (COP) of 2.1 for both heating and cooling [79], the energy consumption for each heating/cooling method to achieve thermal comfort can be calculated. For this, it is considered that the heat pump works for both heating and cooling during the necessary hours of the year (heating when the hourly temperature is below the temperature of comfort and cooling when it is above). Thereby, the energy consumption based upon the hourly difference between the indoor temperature and the temperature of comfort is used, as shown in Equation (2) [80]:

$$EC = \frac{mass \cdot C_p \cdot \left| T_{Comfort} - T_{Passive} \right|}{3600 \cdot COP} \tag{2}$$

In this case, a period of 1 h (3600 s) is used, having hourly values of the energy consumption that are aggregated during an annual period (8760 hours). Thereby, it is considered a regular dwelling of 120 m$^2$ of built area and 2.2 m of height [81], and a temperature of comfort of 22 °C. The specific heat of the air is set at 1.012 kJ/kg°C, and the air density is at 1.2 kg/m$^3$ [82]. With this, and using the heat

pump calculator by Oropeza-Perez [83], Table 9 can be constructed, whereby the energy consumption with the temperatures achieved by the passive methods without any handling is displayed.

**Table 9.** Annual energy consumption by using different passive methods.

| Passive Method | Annual Consumption (kWh) |
|---|---|
| **No passive method** | 1990 |
| **Heating** | |
| Passive solar gain | 1340 |
| Infiltration control | 1380 |
| **Cooling** | |
| Shading system | 1350 |
| Phase change material | 1650 |
| Passive cooling shelter | 1640 |
| Heat sink | 1540 |
| Thermal capacity | 1110 |
| Radiant heat barrier | 360 |
| Eco-evaporative cooling | 1340 |
| Natural ventilation | 1370 |
| Solar-assisted AC | 1410 |
| **Heating/Cooling** | |
| Thermal insulation | 1230 |
| Intelligent facade | 1430 |
| Double-glazed opening | 1430 |

In Table 9, it is clearly seen that the methods without any driving have almost the same amount of energy consumption as is found in a conventional dwelling without any passive method. Therefore, if the improved effectiveness of the methods is taken into account to reach the temperature of comfort, Table 10 can be displayed with the new annual energy consumption.

**Table 10.** Annual energy consumption by using different passive methods and using operation features by the users.

| Passive Method | Annual Consumption (kWh) |
|---|---|
| **No passive method** | 1990 |
| **Heating** | |
| Passive solar gain | 92 |
| Infiltration control | 101 |
| **Cooling** | |
| Shading system | 182 |
| Phase change material | 87 |
| Passive cooling shelter | 59 |
| Heat sink | 105 |
| Thermal capacity | 103 |
| Radiant heat barrier | 13 |
| Eco-evaporative cooling | 18 |
| Natural ventilation | 65 |
| Solar-assisted AC | 36 |
| **Heating/Cooling** | |
| Thermal insulation | 110 |
| Intelligent facade | 41 |
| Double-glazed opening | 105 |

Table 10 shows that there is a considerable reduction of the energy consumption with all methods by having a correct driving of them. The highest consumption (shading systems) is 10 times smaller than the consumption without any passive method, which confirms the necessity of having proper occupant behavior in terms of the control of their own indoor environment.

## 4. Conclusions

An approach to assess the effectiveness of the current passive heating and cooling methods is developed in terms of the indoor temperature increase and decrease, respectively. Based upon the indoor air temperature, and with thermal simulations that calculate the indoor temperature increase/decrease, it is found that, if the methods do not have any driving by the users, the energy consumption could be almost the same as the consumption of a regular dwelling that uses a heat pump to heat and cool the space.

After the proposed simulation modelling, which is located in Mexico City, achieves similar performances of indoor temperature decrease/increase compared with studies of passive methods under similar climate conditions, the results are considered as validated enough to analyze the different driving conditions of the passive methods.

Thus, if some operation feature, defined as mobility, maintenance, assembly, and consumables, is applied to the passive methods, the effectiveness of the passive methods is highly improved in the semi-operable and operable methods, whereas, in those which are defined as not-operable methods (phase change material, thermal insulation, and thermal capacity), the effectiveness remains almost the same. Furthermore, the improvement of the effectiveness is reflected by indoor temperatures that are closer to the temperature of comfort previously set; hence, indoor thermal comfort is achieved.

By applying an integrated driving to the passive methods, the energy saving is significant from not using a heating–cooling heat pump designed to achieved thermal comfort; it was calculated in this document to use at least 10 times less energy than is used by an air-conditioned single dwelling. Hence, it is concluded that a driving by the users is essential to achieve both thermal comfort and energy saving, once the method is installed, and not to assume correct performance without any handling on the part of the user after installation.

**Funding:** This research received no external funding.

**Conflicts of Interest:** The authors declare no conflicts of interest.

## Nomenclature

| | |
|---|---|
| $\varphi$ | effectiveness of the passive methods (0..1) |
| $T_{Passive}$ | indoor temperature achieved by the passive methods (°C) |
| $T_{Comfort}$ | indoor temperature of comfort (°C) |
| $T_{l-u}$ | lower and upper range of thermal comfort (°C) |
| $COP$ | coefficient of performance (dimensionless) |
| $EC$ | energy consumption (kWh) |
| $mass$ | mass of the indoor air (kg) |
| $Cp$ | specific heat of air (kJ/kg·°C) |

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
