# Peer review of "The Influence of an Integrated Driving on the Performance of Different Passive Heating and Cooling Methods for Buildings"

_buildings, doi:10.3390/buildings9110224_

Round 1

Reviewer 1 Report

The manuscript entitled “The influence of an integrated driving on the performance of different passive heating and cooling methods for buildings” aims to analyze the performance of the most common passive heating and cooling methods, in order to estimate their performance in terms of indoor temperature increase/decrease onto a single dwelling.

Overview

First, it should be noted that the writing structure of this article needs to be thoroughly revised. Second, the findings have tested based on the literature, which questions its validity in reality. Third, there is ambiguity about the role of selected case study in this research. Therefore, this article needs substantial improvements to be considered for publication. In following, this reviewer would like to suggest a few comments that authors may consider for improving their work.

Recommendations

[10-12] please start your abstract with a brief overview of the research topic, and then clearly explain state of the art (it looks like this part is missing in your abstract).

[13-16] why in the description of the research methodology, the case of ‘Mexico city’ is missing? If Mexico City is the case study (as stated in line 184), why the validation of findings are based on literature?

[21-23] Please explain the results more clearly. What do the authors mean by general terms of ‘proper usage’ and ‘high influence’?

[49] If the presented image is derived from a source, please cite it.

[54-71] is not it better to summarise this information as a table?

[94] It is highly recommended to include the geographical scope of your study.

[95-175] the methodology section is over-explanation and inappropriate. Authors may transfer much of the data to the prior part (introduction or literature review).

[184] the Mexico City has introduced as the case study, but additional information is needed. How many buildings have considered in this study? What is their location in the city? The direction of the buildings was north-south or south-east? Could please provide a map of studied buildings?

[197] please provide more information about the process of simulation in your research.

[218] could you rephrase your last sentence and assumption about the validity of findings?

[305-311] to consolidate the results, it is suggested that the findings be validated in accordance with the case study in this section.

Author Response

Thank you very much for your useful comments. Your time will make that the manuscript could be improved. The comments were addressed and are highlighted in yellow within the manuscript.

[10-12]: An overview of the research topic as well as a brief state-of-the-art were added on the abstract.

[13-16]: The case study of Mexico City was presented in the abstract. It was stated that the validation was carried with studies with similar climate conditions.

[21-23]: ‘Proper usage’ and ‘high influence’ were further explained.

[49]: The sources of Fig. 1, 4 and 5, were clarified.

[54-71]: The 14 passive methods were gathered on a table.

[94]: The geographical scope of the study was stated.

[95-175]: The description of the passive methods and the features of operation were moved to the Introduction section.

[184]: The information regarding the Mexico City case study was added, including the location within the city, the number of studied buildings and its orientation.

[197]: The simulation was further described.

[218]: The sentences was rewritten according to the fact that the validation of the results relies on the similar climate conditions of the compared studies.

[305-311]: It was stated that the results shown here and in other studies can be considered as similar since the climate conditions are within the same groups (C and D) of the Köppen-Geiger classification system.

Reviewer 2 Report

The study is very interesting and the identification of a guide for end users for a better management of energy control systems is of undoubted interest.

In the paper it would perhaps be better to mention the "case study" in the initial part of the paper - example in the introduction -for better understanding.

Author Response

Thank you very much for your useful comments. Your time will make that the manuscript could be improved. The comments were addressed and are highlighted in green within the manuscript.

“In the paper it would perhaps be better to mention the "case study" in the initial part of the paper- example in the introduction -for better understanding.”

The case study was firstly mentioned in the Introduction section.

Reviewer 3 Report

The article is interesting, it brings new views on the attractive issue of passive heating and cooling. An overview of the literary sources that have been published in this area is formed by an unusual way of dividing into bullets (effects causing various phenomena). The authors present 14 methods influencing heating and cooling, 9 of them cooling. In this document, the above-mentioned passive heating and cooling methods are therefore analyzed in terms of their operating characteristics. The purpose of this is to determine the extent of the decrease / increase in the internal temperature according to the appropriate handling of each passive method and to develop the operation guidance to optimize their performance.
Chapter 2 is clear, the authors divide the method into 4 groups. Formulas (1) but also (2) - missing physical dimension. The temperature is expressed in degrees Celsius, but the physical dimension does not correspond. They are supposed to be Celsius oC not just C. They use the EnergyPus program as a case study using a Mexico city building that has brick walls and a concrete roof. It would be good to describe the building a little bit more.
The results are difficult to check because the input data are not clear. It would be advisable to pay attention to the case study and to describe it in more detail, or to draw pictures, even more clearly for the reader and more attractive.
Conclusion is simple If, ​​therefore, an operating feature, defined as mobility, maintenance, assembly and consumables is applied to passive methods, the efficiency of passive methods is greatly improved on semi-operable and usable methods, while in those defined. Since these are not applicable methods (phase change material, thermal insulation and heat capacity), the efficiency remains almost the same.
There is no need for literature to be sources - numbers with a dot and then in square brackets.
After editing, the article will be attractive in the Buildings magazine, but not in this form. It is still necessary to work on the paper.

Author Response

Thank you very much for your useful comments. Your time will make that the manuscript could be improved. The comments were addressed and are highlighted in blue within the manuscript.

“The temperature is expressed in degrees Celsius, but the physical dimension does not correspond. They are supposed to be Celsius oC not just C.”

The temperature units were changed to °C to meet the physical dimensions.

“They use the EnergyPus program as a case study using a Mexico City building that has brick walls and a concrete roof. It would be good to describe the building a little bit more.”

The building characteristics of construction were further described.

“The results are difficult to check because the input data are not clear. It would be advisable to pay attention to the case study and to describe it in more detail, or to draw pictures, even more clearly for the reader and more attractive.”

The geometry of the simulated dwelling was placed. Nevertheless, due to the lack of space, it was decided not to place pictures of the 14 passive methods. Apologies for this.

“There is no need for literature to be sources - numbers with a dot and then in square brackets”

The arrangement of the literature review was improved.

Round 2

Reviewer 1 Report

Thanks to the authors, the revised version has adequately addressed all the specified parts and has improved significantly. Therefore, I would like to suggest it for publication after a minor revision.

Figure 1: you may replace "Source: own elaboration," with "derived from authors' elaboration."

Table 1: This table should also mention in the context,  "Table 1 presents the most studied and used passive methods..."

Figure 5: This map does not appear to be suitable for an academic paper in the Journal of buildings. If authors would like to use the Google, please be sure that the "labels" are turned off before the screenshot. You can also use other databases like (https://snazzymaps.com/style/104137/default-w-out-labels) to present the map. However, you should cite the used map as a webpage reference. For more information, please see (https://libanswers.aru.ac.uk/faq/78664

Author Response

Once again, thank you very much for your useful comments. Your time will make that the manuscript could be improved. The comments were addressed and are highlighted in yellow within the manuscript.

Figure 1: The source was replaced with “derived from authors’ elaboration” Table 1: The table was mentioned within the manuscript as requested. Figure 5: The map was changed according to your kindly comments and was cited in the references as indicated. Thank you very much for the advice.

Reviewer 3 Report

The article is interesting. The authors corrected what was needed. You can see that they were working on the content. However, it is still necessary to read precisely, to correct deficiencies, eg.

in Fig. 2 ......22 oC

in Fig. 3 ...Y axis .....Inluence of performance....repair Influence

pg. 9 ....roof of 30 cm of with...repair ....of width

You repair literary sources according to the pattern. You write the source-magazine in italics, a year bold.....

You still work on the article, fix bugs. You let it read to someone and fix minor flaws.

Author Response

Once again, thank you very much for your useful comments. Your time will make that the manuscript could be improved. The comments were addressed and are highlighted in blue within the manuscript.

A third reader to avoid bugs reviewed the manuscript.

Caption in Figure 2 was corrected to °C. The axis was corrected to “Influence”. “roof of 30 cm of with” was corrected to “roof of 30 cm of width”. Literature sources were adjusted to the Journal requirements.